# Post-Traumatic Growth and Resilience among Hospitalized COVID-19 Survivors: A Gendered Analysis

**DOI:** 10.3390/ijerph191610014

**Published:** 2022-08-14

**Authors:** Samuel Adjorlolo, Paul Adjorlolo, Johnny Andoh-Arthur, Emmanuel Kwadzo Ahiable, Irene Akwo Kretchy, Joseph Osafo

**Affiliations:** 1Department of Mental Health, School of Nursing and Midwifery, College of Health Sciences, University of Ghana, Legon, Accra P.O. Box LG 43, Ghana; 2Research and Grant Institute of Ghana, Legon, Accra P.O. Box LG 1004, Ghana; 3Department of Statistics and Actuarial Science, College of Basic and Applied Sciences, University of Ghana, Legon, Accra P.O. Box LG 43, Ghana; 4Department of Psychology, School of Social Sciences, College of Humanities, University of Ghana, Legon, Accra P.O. Box LG 84, Ghana; 5The Greater Accra Regional Hospital, Ridge, Ghana Health Service, Accra P.O. Box 473, Ghana; 6Department of Pharmacy Practice and Clinical Pharmacy, School of Pharmacy, College of Health Sciences, University of Ghana, Legon, Accra P.O. Box LG 43, Ghana

**Keywords:** resilience, post-traumatic growth, COVID-19, SARS-CoV-2, gender, Africa

## Abstract

The literature on behavioral outcomes associated with the COVID-19 pandemic is inundated with mental health burdens such as depression and stress disorders. The current study investigated gender invariance on resilience and post-traumatic growth (PTG) as positive psychological changes associated with the COVID-19 pandemic. A total of 381 survivors of the COVID-19 infection completed measurements of resilience, PTG, violence and stigma experience, and mental health problems like post-traumatic stress disorder (PTSD). The data were analyzed using descriptive statistics, correlation, multivariate regression, and a latent profile analysis. The results revealed that more than half of the participants had high scores on resilience (53.6%) and PTG (60.9%). The positive psychological changes, although independent of each other, were moderated by gender, and influenced by the negative experiences of participants such as stigma, violence, and PTSD. Latent profile analyses revealed three classes of participants, two of which were characterized by high scores on mental health problems and PTG. The clusters were invariant across gender. Surviving COVID-19 contributed to resilience and PTG. These can be targeted for intervention programs to mitigate the mental health burden occasioned by the pandemic.

## 1. Introduction

Since the emergence of the Coronavirus disease 2019 (COVID-19) in Wuhan, China, and its subsequent declaration as a pandemic by the World Health Organization [1], the research community has shown profound interest in explaining the impact of the pandemic on humanity. To date, the literature is inundated with the adverse changes induced by the COVID-19 pandemic. Specific to mental health, the high burden of mental health problems, such as depression, anxiety and post-traumatic stress disorder (PTSD) have been reported in diverse populations, including children and adolescents [2,3,4], the general population [5], healthcare workers [6] and patients diagnosed with COVID-19 [7]. When stratified by gender, the evidence is mixed, with some studies reporting more mental health problems among females [8], while others found no gender difference [7]. This stream of research has been useful in increasing our understanding and informing policy programs and interventions by governments and regional bodies to ameliorate population mental health [9].

However, decades of research have shown that not all individuals exposed to adversities develop mental health problems. Some of these individuals tend to develop positive emotional states and growth, often referred to as positive psychological health [10,11,12,13]. Among these changes are post-traumatic growth (PTG) and resilience [11,14,15,16]. In an earlier research, Bonanno challenged the dominant grief assumption that repeatedly highlights the symptoms of distress such as depression and post-traumatic stress disorder (PTSD) as common reactions to disasters [11]. This is not to suggest that individuals who experience resilience and PTG are shielded from manifesting negative symptoms associated with pandemics such as COVID-19; rather their daily lives are not compromised significantly by the pandemic [16]. The over concentration on mental health symptoms, particularly among COVID-19 survivors, not only underestimates the human capacity to thrive after adverse events [11] but also deprives the research community of the insight into the positive psychological changes associated with the COVID-19 pandemic [17,18,19,20]. Understanding the extent and nature of positive psychological health associated with the COVID-19 pandemic would support comprehensive intervention programming, including supportive behavioral changes to promote a healthy psychological state [21]. The study investigated PTG and resilience among COVID-19 survivors in Ghana, a sub-Saharan African country.

## 2. Overview of Resilience and Post-Traumatic Growth

The resilience construct, also referred to as “stress resistance” and “invulnerability”, has been studied from different disciplinary perspectives, including psychology, psychiatry, ecology and physics [22]. Defined variously, resilience is operationalized as the ability to adapt positively or maintain adequate functioning in the face of adverse conditions [11,23,24]. As opposed to recovery, resilience is not merely the absence of psychopathology but a reflection of the ability to maintain a stable equilibrium in the face of adversities [11]. Resilient individuals have the capacity for positive experiences and emotions, and are able to delay the experiences of disruptions in their daily functioning [25]. Research has uncovered the active ingredients in resilience to include hardiness, self-enhancement, repressive coping, positive emotion, and laughter. The ingredient of hardiness, for instance, encompasses an individual belief to influence the immediate surrounding and event outcomes, the ability to learn from both negative and positive experiences and a commitment to finding meaningful purpose in life [11]. PTG, on the other hand, encapsulates the tendency of an individual to witness significant positive changes following exposure to a major life crisis [26]. The development of PTG, following adversities, is underpinned by a myriad of processes such as reframing experiences, focusing on the perceived positive benefits from life trauma [27], adjustments to life priorities, and the adoption of a healthy lifestyle [28]. These PTG pathways strategically position individuals to improve their relationships with others, create new possibilities, advance personal strength, experience spiritual change, or increase the appreciation of life [27].

## 3. COVID-19 Pandemic, Resilience and Post-Traumatic Growth 

COVID-19 is a global public health emergency and disaster that is characterized by a high sense of insecurity occasioned by high infection and death rates. While the association between COVID-19 and mental health burden is unsurprising, the interesting question is whether COVID-19 can induce positive psychological changes. Researchers have addressed this seemingly important question by focusing on the working population, such as healthcare workers [16,29,30,31]. Finstad et al. synthesized 46 articles on positive changes induced by the COVID-19 pandemic in the workplace, of which 34 were on resilience, PTG, and coping strategies [16]. The results showed that resilience correlated positively with PTG, life and professional satisfaction, participant’s age and level of education, perceived social support, and sleep quality, but negatively with work-related stress, anxiety, burnout, and depression. On PTG, Finstad et. al. observed that frontline healthcare workers appeared to have exhibited higher levels of PTG. Importantly, PTG was found to be dependent on several factors, such as self-confidence, awareness of risk of infection, psychological support, adaptive coping strategies, resilience, and positive reappraisal of events.

The concentration on frontline workers, including healthcare professionals, was mainly because they can render optimal care and support only if they stay healthy psychologically and physically. At the same time, individuals who have recovered from COVID-19 are resourceful agents as they navigate the sociocultural context that define and shape responses and ultimately their psychological and emotional states. However, only a few studies have focused on resilience and PTG among COVID-19 survivors [17,18,19,20,32]. Among Chinese participants, resilience has been found to correlate significantly and negatively with anxiety, depression, and PTSD [18], and mediated the associations between stigma and sleep quality [19]. Among Mexican samples, resilience was found to significantly moderate the expression of psychopathological behaviors such that higher resilience buffered individuals from expressing high symptoms of depression, anxiety and stress [32]. Relating to PTG, a study involving Chinese COVID-19 survivors found that PTG was facilitated by a reassessment of life priorities, goals, and values and appreciation of life [17]. Similar findings have been reported among COVID-19 survivors in Turkey [33]. In another study involving Chinese COVID-19 survivors, social support, self-stigma, receiving mental health care services during hospitalization, self-esteem, PTSD, and coping strategies significantly predicted PTG [20,34]. 

## 4. The Current Study

As discussed, limited attention has been granted to resilience and PTG among COVID-19 survivors. These individuals, by virtue of their experiences with the various phases of COVID-19, are strategically positioned as a vital source of data on the dynamics and characteristics of resilience and PTG to support discourses on the ramifications of COVID-19. Therefore, research focusing on their positive psychological growth is timely and important. Second, gender has a major influence on the prevalence, severity, clinical outcomes, and other characteristics of COVID-19, with the bulk of the evidence pointing towards gender variance [35,36,37]. For example, data from 1000 patients from China revealed that COVID-19 severity and death rates were significantly higher in males then females [38]. In another study, males were significantly more likely to be hospitalized, transferred to intensive care unit, received vasopressor support, and endotracheal intubation [39]. We reasoned the reported gender differences in COVID-19 would influence not only the development of mental health problems [7,8] but also resilience, PTG, and other positive psychological behaviors. This makes it extremely important to elucidate the extent to which resilience and PTG associated with COVID-19 are invariant across gender. Consequently, the current study contributed to decolonizing the COVID-19 literature by conducting gender-sensitive analyses to achieve the following objectives:

The first objective was to investigate the prevalence of resilience and PTG among COVID-19 survivors and the predictors of resilience and PTG. Previous studies have suggested that negative outcomes such as stigmatization and positive experiences such as social support contribute to the formation of resilience and PTG experiences [20,34]. In line with the above, the second objective of the study was to investigate the following as predictors of resilience and PTG: stigma, violence victimization, PTSD, psychological distress, sleep difficulty, and social support. Lastly, we investigated whether there exist latent subpopulations within the COVID-19 survivor population. That is, whether there are distinct categories of COVID-19 survivors based on the predictors and outcome variables. This objective was inspired by the observation that people who share the same experiences or encounters are not necessarily a homogenous group, and that they could have different configural profiles based on a set of personal and/or environmental attributes [40]. This would add an additional layer of evidence to the characteristics of COVID-19 survivors. 

## 5. Methodology

*Participants*: Data were collected from individuals diagnosed with COVID-19 who were hospitalized and/or received treatment for COVID-19 at an outpatient department (OPD). Operationalized as COVID-19 survivors, they were recruited for a longitudinal cohort study designed to investigate possible changes in the experience of the negative and positive psychological changes associated with the COVID-19 diagnosis and treatment. A total of 381 COVID-19 survivors completed the first wave of data collection. They were recruited from the healthcare facilities in the Greater Accra Region of Ghana designated for COVID-19 treatment. Almost half of the participants (*n* = 186, 48.8%) were recruited from Ga East Municipal Hospital (GEMH), which was designated specifically to treat and manage only COVID-19 cases during the early days of the pandemic in Ghana. This was followed by the Ghana Infectious Disease Center (GIDC; *n* = 161, 42.3%). GIDC was built as Ghana’s premier infectious disease center in the wake of the COVID-19 pandemic. Lastly, 34 (8.9%) participants were recruited from the Greater Accra Regional Hospital (also known as Ridge Hospital). 

*Research Design*: This was a prospective longitudinal cohort study involving the collection of data from the same participants over a 12-month period. A four-wave sequential mixed-method data collection consisting of administration of questionnaires and individual in-depth interviews (IDIs) collection was planned, with 3-month intervals between the waves. This study reports on the quantitative data generated in the first wave of data collection.

*Procedure for Data Collection*: The study commenced at the time that most of the COVID-19 patients had been discharged, thus making it difficult for participant recruitment at the time of discharge, as originally planned. In the revised scheme of work, the management of GEMH and Ridge Hospital invited the participants for the study. The invitation involved a brief overview of the study and consent for the contact information of those interested in the study to be given to the research team. A follow-up invitation was extended to the participants to form the study cohort. To establish the rapport necessary for a longitudinal cohort study, the participants were invited to the health facilities where they were treated and discharged for initial, face-to-face interaction with the research team. The hospital management provided a dedicated office, spacious enough to accommodate a social distancing protocol for the data collection. Each participant was scheduled accordingly based on their earliest convenient time. The research team provided reminders in the form of text messaging and phone calls to support the participants to adhere to the appointment schedules. Unlike Ridge Hospital and GEMH, participants from GIDC were recruited at the OPD where they were scheduled for a review after discharge.

On the day of data collection, the participants were given the informed consent form. They were asked to review the information provided carefully. They were encouraged to ask any question bothering them to ensure that they were satisfied and understood the study objectives and their responsibilities as participants before consenting. Ethical issues such as confidentiality, anonymity, voluntary withdrawal, and protection from harm were extensively discussed with the participants to build their trust in the research. For those who could not read the English language, the consent form was provided in the local language of their choice. Interpreters and Ghanaian language experts were readily available to render support to the participants. Upon registering their consent, they completed a participant identification form, after which a unique identification code was generated for each participant to assist in a follow-up and matching of data. The identification codes were linked to the participants’ data, namely name and contact details in an Excel spreadsheet, which is password protected to prevent unauthorized access to the participants’ data and identifying information. Next, the participants were handed a questionnaire pack containing the measures described below. The completed questionnaires were handed over to the research team. The participants were reimbursed for the transportation cost and time spent participating in the study. All the approved COVID-19 risk reduction and mitigation protocols were adopted such as handwashing under running water, the use of alcohol-based hand sanitizers. Some members of the research who were unwell underwent testing for COVID-19 as part of the precautionary measures. 

## 6. Data Collection Measures

*The Brief Resilience Scale* [41] was used to assess resilience. Three items judged to have face validity and applicable to the local context were used to index resilience due to COVID-19. The items were rated on a 5-point response scale ranging from 1 (strongly disagree) to 5 (strongly agree). A higher score indicates a higher degree of resilience. The Cronbach’s alpha reported in this study was 0.62.

*Posttraumatic Growth Inventory Short Form* (PTGI-SF) is a 10-item scale [42] derived from the 21-item Post-Traumatic Growth Inventory [12]. The PTGI-SF measures growth in areas such as relating to others, new possibilities, personal strength, spiritual change, and appreciation of life. The items are rated on a 6-point Likert scale from 0 (no change) to 5 (very great degree of change). The following subscales were used in this study given that they appeared relevant to COVID-19: personal strength, spiritual change, and appreciation of life. A Cronbach alpha of 0.75 was reported in the study.

*Patient Health Questionnaire-4* was used to assess symptoms of psychological distress. The PHQ-4 consists of the PHQ-2 [43] and 2 items from the Generalized Anxiety Disorder (GAD) scale [44]. The PHQ-4 reflects the core diagnostic criteria for both major depressive disorder and anxiety disorder contained in the Diagnostic and Statistical Manual of Mental Disorders-IV [45]. The PHQ-4 response options ranged from “not at all” (0) to “nearly every day” (3), with high scores reflecting more psychological distress. A Cronbach alpha of 0.80 was reported in the study.

*The COVID-19 Stigma Scale* (CSS) was adapted from the existing 12-item HIV Stigma Scale short-form version [46] developed to assess stigma, including personalized/enacted stigma, perceived concerns about public attitudes, and negative self-image. The study focused on personalized/enacted stigma and perceived concerns about public attitudes, as it was reasoned that they appeared more relevant in the context of COVID-19 [47]. The scale was modified by indicating “coronavirus” or “diagnosed and treated for coronavirus” in place of HIV. The scale’s item was each rated on a four-point scale from “strongly disagree” (1) to “strongly agree” (4), with the total score obtained by summing the respective items. The scale’s internal consistency (i.e., Cronbach’s alpha) was 0.79. 

*The Post-Traumatic Stress Disorder scale* was used to assess symptoms of PTSD. The scale consists of 5 items derived from the validated PTSD-8 [48] designed to reflect the DSM-IV criteria for PTSD. The PTSD scale items were answered on a five-point Likert scale, ranging from 0 (never) to 5 (always). The summed score relates to symptom severity, with higher scores indicating more symptoms of PTSD. In this study, a Cronbach’s alpha of 0.72 was reported.

*The COVID-19 Victimization Scale* was developed to assess victimization experiences associated with diagnosis of COVID-19. The scale was developed following a review of the violence and crime literature. The items constituting the scale were extracted from existing studies that have measured violence in the community [49,50]. The 6-item scale measures both psychologically and physically violent experiences and was scored using a five-point Likert scale that ranges from 0 (never) to 5 (always). A Cronbach’s alpha of 0.81 was recorded for the COVID-19 Victimization Scale.

*Sleep Difficulty Scale*: The sleep difficulty associated with COVID-19 was measured by asking about (1) difficulty to fall asleep while in bed and (2) difficulty to stay asleep through the night. The items were extracted from the existing literature [51] and were scored using a four-point Likert response format ranging from not at all (0) to very often (3). Responses to each item were added to create a total score, with higher scores indicating more sleep difficulty. The Cronbach’s alpha for the sleep difficulty scale was 0.89. 

*The COVID-19 Social Support Scale* was extracted from the Medical Outcomes Study Social Support Survey (MOS-SSS) [52] as a multidimensional measure of social support. For brevity, 5 items that are contextually relevant were extracted from the MOS-SSS and were scored on a 5-point Likert response from 0 (never) to 5 (always). The COVID-19 Social Support Scale recorded a Cronbach Alpha of 0.88. 

## 7. Data Analytic Strategy

Descriptive statistics were used to provide a summary of the demographic data as well as the extent of resilience and PTG in the participants. To estimate prevalence, the mean scores of resilience and PTG were converted into standardized scores, with a mean of 0 and a standard deviation of 1. Scores below and above the mean were designated as low and high, respectively, with the latter reflecting the prevalence rate. A mean-level analysis was conducted using the independent sample *t*-test and analysis of variance (ANOVA) to examine demographic differences on resilience and PTG. Intercorrelations between the study variables were investigated using Pearson product-moment correlation test while a multiple linear regression was conducted to investigate the predictors of resilience and PTG. Analyses were performed using the IBM SPSS Statistics for Windows version 26, Armonk, NY, USA: IBM Corp.

We also conducted a latent profile analysis (LPA) to determine whether the participants form a distinct subgroup based on their scores in the study variables. LPA, along with other latent class clustering analyses such as the latent transition analysis, aimed to group participants based on their similarities on latent variables that were not measured directly but were inferred based on a set of measured variables. We estimated models with 1 (Model 1), two (Model 2), and 3 (Model 3) latent profiles to determine the optimal latent profile model. Each model was examined, and the best model selected based on entropy and goodness of fit indices. Entropy measures the predictive ability of the various models or model distinctiveness, with values ranging from 0 to 1; high values indicate a better class membership prediction [53,54]. The Akaike Information Criteria (AIC) and Bayesian Information Criteria (BIC) model fit indicators were used to compare the three models estimated. Lower AIC and BIC indices indicated a better model fit over the successive models. LPA was conducted with R-software version 4.1.3, using the robust maximum likelihood [53]. 

## 8. Results

### 8.1. Demographic Characteristics of Participants

The demographic characteristics of the participants are summarized in Table 1. The average age of the participants was 43.10 (SD = 15.62). Males constituted more than 50% of the sample. Participants who completed post-secondary school education (i.e., more than 12 years of formal education) were over-represented. Most of the participants indicated that religion is important in their life (95.8%). 

### 8.2. Prevalence of COVID-19-Induced Resilience and Post-Traumatic Growth

On average, more than half of participants agreed or strongly agreed that COVID-19 has impacted positively on their resilience (Table 2). For example, 65.7% of the participants indicated that it took them a short time to overcome the setbacks in their lives occasioned by the COVID-19 pandemic. Similarly, a large percentage of participants agreed or strongly agreed that they experienced growth after recovering from COVID-19. About 82.6% of the participants agreed or strongly agreed that COVID-19 had facilitated the appreciation of the value of their lives. The overall prevalence was obtained by aggregating “Agree” and “Strongly Agree” responses. In terms of overall prevalence, more than half of the participants experienced resilience (60.9%) and PTG (53.6%) due to COVID-19. The mean-level analysis revealed that resilience among males (M = 10.43, SD = 2.78) was significantly higher than in their female counterparts (M = 9.86, SD = 2.56), t (377) = 2.02, *p* < 0.05. There was a significant statistical difference in males (M = 21.68, SD = 5.38) and females (M = 23.10, SD = 4.53) on PTG, t (375) = −2.68, *p* < 0.01, suggesting that females experienced more PTG. Significantly high resilience was reported by participants with high education (M = 10.36, SD = 2.63), relative to those with low education (M = 9.71, SD = 2.85), t (377) = −2.03, *p* < 0.05. Likewise, participants who rated religion as important (M = 22.38, SD = 5.03) reported significantly higher PTG than those who scored low on the importance of religion (M = 19.33, SD = 5.83), t (375) = −2.29, *p* < 0.01. 

### 8.3. Intercorrelations between the Study Variables

The results of the correlation analysis, separated by gender, are summarized in Table 3. Among males, resilience correlated significantly with adverse experiences such as stigma (*r* = −0.24, *p* << 0.01), PTSD (*r* = −0.20, *p* < 0.01), and psychological distress (*r* = −0.20, *p* < 0.01). In females, however, resilience correlated with violence experience (*r* = −0.17, *p* < 0.05). With respect to PTG in males, PTSD (*r* = 0.32, *p* < 0.01), psychological distress (*r* = 0.26, *p* < 0.01), and sleep difficulty (*r* = 0.32, *p* < 0.01) emerged as significant correlates. Among females, PTG was significantly correlated with PTSD (*r* = 0.25, *p* < 0.01) and social support (*r* = 0.27, *p* < 0.01). There was also evidence of gender differences in some correlation patterns. For example, resilience and stigma were negatively correlated for males (*r* = −0.24, *p* < 0.01); for females, however, there was no significant correlation.

### 8.4. Predictors of Resilience and Post-Traumatic Growth by Gender 

In a multivariate regression model in which other variables were controlled, the results (Table 4) showed that stigma significantly predicted resilience in males (*β* = −0.09, *p* < 0.01). Violence experience (*β* = −0.36, *p* < 0.05), PTSD (*β* = 0.32, *p* < 0.01) and psychological distress (*β* = 0.20, *p* < 0.05), were significant predictors of PTG in males. Among females, however, only social support emerged as a significant predictor of PTG (*β* = 0.46, *p* < 0.01).

### 8.5. Cluster Membership of Participants

A latent profile analysis revealed that a 3-latent cluster provided an adequate fit to the data (AIC = 14,875.99, BIC = 15,010.05, Entropy = 0.97) as opposed to 2-latent (AIC = 15,068.26, BIC = 15,166.83, Entropy = 0.98) or 1-latent clusters (AIC = 15,469.77, BIC = 15,532.85, Entropy = 1). Univariate ANOVAs revealed that the three cluster groups differ significantly on all the nine variables (all *p* ˂ 0.001). Membership in cluster 1, 2, and 3 are distributed as 136 (36.56%), 53 (14.25%), and 183 (49.19%), respectively, with a one-way multinomial chi-square test revealing that the frequencies are not distributed equally [χ^2^ (2) = 69.89, *p* < 0.001)]. As shown in Table 5, participants in cluster 1 were defined primarily by the experience of relatively high stigma, followed by PTSD and violence but low social support. The participants in cluster 2 experienced relatively high levels of suicidal behaviors, psychological distress, and PTSD, followed by moderately high sleep difficulty, violence, and stigma. They also reported increased levels of PTG but were rather low in resilience. Lastly, the participants in cluster 3 scored relatively low on the psychopathological behaviors (i.e., psychological distress and PTSD), stigma, violence experience, suicidal behaviors, and PTG. They had moderately high social support and resilience scores. Further analysis using chi square revealed that the cluster membership was not significantly influenced by gender [χ^2^ (2, *n* =372) = 4.87, *p* = 0.088)].

## 9. Discussion

Traumatic events, including infectious disease epi/pandemics do not only contribute to mental health problems but also to positive psychological changes. To contribute to the burgeoning cross-culture literature on COVID-19-induced positive psychological changes, the current study investigated resilience and PTG among participants diagnosed and treated for COVID-19 in Ghana, a sub-Saharan African country.

It was observed that positive psychological experiences occasioned by the COVID-19 pandemic were fairly distributed among the study participants. More than half of the participants reportedly experienced resilience and PTG; however, slightly more participants experienced resilience, compared with PTG. A gendered analysis revealed that males were more resilient than their female counterparts, whereas females experienced significantly more PTG than males. Resilience in males could be accounted for by the gendered socialization practices in Ghanaian societies that are intended to inculcate the traits of “hardiness” and “toughness” in males to withstand adversities [55]. This view largely resonates with the popular saying in Ghana that a “man does not cry or shed tears”, which implies that a man must be resilient and resolute in the face of calamities without or with minimal outward display of psychological and emotional breakdown. Gender socialization may thus account for the high resilient scores in males even though it has been documented that such gendered experiences and the expression of health and wellbeing among Ghanaian males could also undermine actual health and wellbeing leading to negative outcomes in males [56]. Females, on the other hand, are socialized as vulnerable individuals who are to be cared for, supported, and protected from harm [55]. Thus, when faced with adversities, females are more likely to gather support and resources pertinent to experiencing new growth. Another plausible explanation relates to differences in religious engagement between females and males, which has been found to contribute to PTG in this and previous studies [57]. Females are deeply committed to and routinely engaged in religious activities, relative to males [58,59,60]; an observation that could explain females experience of PTG in this study.

Varying relationships have been observed between resilience, PTG, and mental health behaviors, consistent with the existing literature. Of particular interest is the relationship between resilience and PTG which appears complicated in the literature, with some studies reporting no significant relationship [61,62,63], while others have found positive [64,65], negative [66,67], and curvilinear relationships [68,69]. Lending partial support to the literature, our finding shows that resilience and PTG can act independently of each other, and that PTG is not necessarily dependent on resilience, nor is resilience a pre-condition for PTG. Resilience, as a personal resource or growth condition, can predate or occur due to trauma. PTG, on the other hand, is a by-product of traumatic events that bothers on transformational changes produced by challenging an individual’s core beliefs held prior to the traumatic event [61,70]. For example, an individual would alter their views on social support after receiving and recognizing the importance of social support after a COVID-19 diagnosis. This positive psychological change can occur regardless of an individual’s resilience, and it is largely dependent on previous experiences and how an individual successfully overcomes traumatic events by deploying the necessary resources to adapt and adjust to the stress-producing events. This level of distinctiveness creates the possibility for resilience and PTG to exist independently following traumatic events, as well as to contribute to different outcomes such as experience of depression and other negative psychological states as reported in this and previous studies [57]. 

In another point of discussion, the results of the cluster analyses revealed that survivors of COVID-19 are not necessarily homogeneous in terms of their experience of mental health problems, violence, stigma, and positive psychological growth. Rather, there is a possibility of heterogeneous groups of participants who may need different support systems, including intervention programming. The participants in the first two clusters were defined primarily by their relatively high experience of mental health problems, stigma, and violence but with low social support and resilience. In contrast, cluster 3 participants registered relatively low mental health problems, stigma, violence, and PTG, but relatively high resilience. Further analyses revealed that cluster membership did not differ across gender, suggesting that males, as well as females could belong to any of the clusters. 

The results of the multivariate regression analysis appear to suggest gender variance regarding the predictors of PTG and resilience. For instance, only stigma significantly predicted resilience in males. Relatedly, the PTG among males was negatively influenced by violence experience, but positively by PTSD and psychological distress. Thus, a high level of COVID-19-related psychological stress motivated PTG among males [71]; however, this can be truncated by the experience of violence, which in this study was indexed predominantly by physical violence. Among females, however, social support emerged as the only significant contributor to PTG. Precisely, females who received social support reportedly experienced more PTG. As noted previously, social support is a key resource among females. In Ghanaian cultures, females have the tendency to willingly and readily accept support systems, relative to males who are supposedly socialized to be tough and hardy to withstanding adversities [55].

## 10. Implications for Interventions

Individuals who have recovered from COVID-19 infection are resourceful agents as they navigate the sociocultural context that defines and shapes responses to the pandemic. Insight into their experiences could be added to the gamut of programs and interventions designed to expand public education and prevention of COVID-19. We found evidence for positive psychological changes, namely PTG and resilience following COVID-19 diagnosis and treatment, in addition to mental health problems, stigma, and violence. However, these positive changes appeared orthogonal to each other. Thus, an individual who appeared resilient may not necessarily experience PTG and vice versa, regardless of gender. Given the contributions of resilience and PTG in recovery and survival of traumatic events, a systematic effort to screen COVID-19 patients for these constructs is ultimately necessary to inform evidence-based decision making regarding support systems. Consistent with previous studies, resilience may potentially play a preventive role in the development of stigma and could be an important component of anti-stigma interventions [72], particularly among males.

Likewise, efforts to reduce violence experience in males would enhance their PTG following COVID-19 diagnosis and treatment. Among females, however, efforts to increase the social support system would inure significantly to their PTG experiences.

## 11. Study Limitations

The study findings should be reviewed considering the following limitations. The use of self-report measures or questionnaires does not provide an objective means of verifying the accuracy of the information presented. There is, therefore, the possibility that the responses of the participants are under/overreported. Second, the data collection process required the participants to recollect their experiences following discharge from the hospital. Challenges associated with retrospective recall such as memory decay could compromise the experiences reported by the participants. While it is a common practice to obtain data from collateral sources such as significant others, the behaviors of interest to the study are those that can be best represented by the participants. The participants who agreed to participate in the study could be self-selecting and so may be demographically different from those who refused participation, thereby limiting the generalization of study findings to survivors of COVID-19. Time after a traumatic event is extremely important in PTG. The study findings relating to PTG are limited by the lack of inclusion of time sensitive measures such as duration of treatment and after discharge. Lastly, data were collected after the country had experienced wave two of the COVID-19 pandemic. Compared with the early phase of the pandemic, the negative experiences associated with a COVID-19 diagnosis appeared to decrease with time, particularly as the death rate from the virus was extremely low. This development could also affect community responses to COVID-19 survivors. 

## 12. Conclusions

The current study has added to the repertoire of studies investigating positive psychological changes associated with the COVID-19 pandemic. The findings have contributed to the mental health and COVID-19 literature, which currently are predominantly deficit oriented with a focus largely on mental health problems related to the COVID-19. COVID-19 diagnosis and treatment have the tendency to contribute to an individual’s resilience and PTG. Individuals diagnosed with COVID-19 have been ushered into a “new world” where they have the opportunity to question their beliefs, practices and other behavioral tendencies that primarily define their interaction with the world. In doing so, these individuals would appear well positioned to accept activities that contribute to their recovery and reformulate new life goals and expectations that incorporate the experiences associated with the COVID-19 diagnosis and treatment trajectory. These positive psychological changes can be independent of each other, partly moderated by gender, and can be contributed to by the negative experiences encountered by the participants, such as stigma, violence experience, and PTSD. Although gender did not moderate the clustering of the participants, the findings of the study point to the importance of designing gender-sensitive interventions to improve resilience and PTG. Future studies should explore in detail the mechanisms of development of resilience and PTG. 

## Figures and Tables

**Table 1 ijerph-19-10014-t001:** Demographic Characteristics of Participants.

	Frequency (*n*)	Percentage
**Marital Status**		
Single	115	33.7
Married	197	57.8
Other	29	8.5
**Gender**		
Male	226	59.3
Female	155	40.7
**Level of Education**		
Low	94	24.7
High	287	75.3
**Religion**		
Christian	356	93.4
Other	25	6.6
**How important is religion in your life**		
Not Important	16	4.2
Important	365	95.8
**Is your family aware that you have contracted COVID-19**		
Yes	356	93.4
No	25	6.6
**Is your local community aware that you contracted COVID-19**		
Yes	86	22.6
No	295	77.4

**Table 2 ijerph-19-10014-t002:** Prevalence of Resilience and Post-Traumatic Growth Experiences.

	Strongly Disagree	Disagree	Neutral	Agree	Strongly Agree	≤Neutral	≥Agree
COVID-19 Induced Resilience	*n* (%)	*n* (%)	*n* (%)	*n* (%)	*n* (%)	*n* (%)	*n* (%)
It does not take me a long to recover from COVID-19 event.	29 (7.7)	82 (21.6)	50 (13.2)	119 (31.4)	99 (26.1)	161 (42.5)	218 (57.5)
I usually come through difficult times involving COVID-19 with little trouble	52 (13.7)	92 (24.3)	50 (13.2)	131 (34.6)	54 (14.2)	194 (51.2)	185 (48.8)
I took a short time to get over setbacks in my life due to COVID-19	24 (6.2)	59 (5.6)	47 (12.4)	156 (41.2)	93 (24.5)	130 (34.3)	249 (65.7)
Overall prevalence						149 (39.1)	230 (60.9)
**COVID-19 Induced Post-Traumatic Growth**							
I changed my priorities about what is important in my life after recovering from COVID-19.	75 (19.8)	80 (21.1)	19 (5.0)	121 (31.9)	84 (22.2)	174 (45.9)	205 (54.1)
I have a greater appreciation for the value of my own life after recovering from COVID-19.	17 (4.5)	42 (11.1)	7 (1.8)	145 (38.2)	168 (44.3)	66 (17.4)	313 (82.6)
I discovered that I am stronger than I thought after recovering from COVID-19.	22 (5.8)	58 (15.3)	31 (8.2)	144 (38.0)	124 (32.7)	111 (29.3)	268 (70.7)
I know better that I can handle difficulties after recovering from COVID-19.	16 (4.2)	42 (11.1)	27 (7.1)	166 (43.9)	127 (33.6)	85 (22.5)	293 (77.5)
I have a better understanding of spiritual matters after recovering from COVID-19	37 (9.8)	54 (14.2)	54 (14.2)	121 (31.9)	113 (29.8)	145 (38.2)	234 (61.7)
I have a stronger religious faith after recovering from COVID-19.	27 (7.1)	60 (15.8)	33 (8.7)	113 (29.7)	147 (38.7)	120 (31.6)	260 (68.4)
Overall prevalence						175 (46.4)	305 (53.6)

**Table 3 ijerph-19-10014-t003:** Correlations among the Study Variables.

	Variables	1	2	3	4	5	6	7	7	9	10
1	Resilience	1	0.01	−0.24 **	−0.12	−0.20 **	−0.20 **	−0.08	0.07	−0.14 *	−0.10
2	Post-traumatic growth	0.08	1	0.02	−0.03	0.32 **	0.26 **	0.21 **	0.03	0.07	0.22 **
3	Stigma	−0.04	0.04	1	0.15 *	0.17 *	0.19 **	0.13	−0.18 **	0.07	0.13
4	Violence Experience	−0.17 *	0.04	0.32 **	1	0.11	0.15 *	0.13	0.03	0.22 **	0.21 **
5	PTSD	0.03	0.25 **	0.34 **	0.30 **	1	0.47 **	0.39 **	−0.10	0.25 **	0.24 **
6	Psychological distress	0.04	0.16	0.28 **	0.22 **	0.55 **	1	0.30 **	−0.07	0.20 **	0.15 *
7	Sleep difficulty	0.02	0.15	0.05	0	0.31 **	0.32 **	1	−0.08	0.14 *	0.24 **
8	Social support	0.07	0.27 **	−0.16 *	−0.23 **	0.03	−0.09	0.07	1	0.06	0.03
9	Help from MH professionals	−0.16 *	0.05	0.23 **	0.26 **	0.24 **	0.18 *	0.28 **	0	1	0.30 **
10	Help from spiritual leaders	0.03	0.14	0	0.15	0	0.16 *	0.08	0.06	0.32 **	1

Note: PTSD = Post-traumatic stress disorder, MH = Mental health, Correlations above the diagonal = males; Correlations below the diagonal = females, *p* < 0.01 **, *p* < 0.05 *.

**Table 4 ijerph-19-10014-t004:** Multiple regression among study variables.

	Resilience				Post-Traumatic Growth			
	Males		Females		Males		Females	
Variables	Β (SE)	*t*	Β (SE)	*t*	Β (SE)	*t*	Β (SE)	*t*
**Educational level**	0.01 (0.45)	0.03	0.81 (0.49)	1.66	−2.06 (0.83)	−2.48 **	−0.26 (0.81)	−0.33
**Importance of religion**	−0.04 (0.78)	−0.05	−0.99 (1.93)	−0.51	3.37 (1.43)	2.36 *	3.27 (3.17)	1.03
**Stigma**	−0.09 (0.03)	−2.78 **	−0.02 (0.03)	−0.41	−0.05 (0.06)	−0.82	0.01 (0.06)	0.19
**Violence Experience**	−0.09 (0.10)	−0.94	−0.16 (0.10)	−1.59	−0.36 (0.18)	−2.02 *	0.07 (0.16)	0.44
**PTSD**	−0.10 (0.06)	−1.81	0.03 (0.06)	0.51	0.32 (0.10)	3.18 **	0.18 (0.10)	1.75
**Psychological Distress**	−0.07 (0.05)	−1.26	0.03 (0.06)	0.54	0.20 (0.10)	2.00 *	0.09 (0.10)	0.87
**Sleep Difficulty**	0.05 (0.09)	0.61	−0.02 (0.10)	−0.22	0.27 (0.16)	1.65	0.19 (0.16)	1.21
**Social Support**	0.01 (0.06)	0.17	0.01 (0.07)	0.09	0.13 (0.12)	1.11	0.46 (0.12)	4.00 **
	R² = 0.10		R² = 0.06		R² = 0.20		R² = 0.17	
	F (8213) = 2.94		F (8144) = 1.10		F (8211) = 6.15		F (8144) = 1.10	
	*p* < 0.01				*p* < 0.01		*p* < 0.01	

*p* < 0.01 **, *p* < 0.05 *.

**Table 5 ijerph-19-10014-t005:** Cluster Profile and Gender Distribution across Clusters.

	Cluster 1	Cluster 2	Cluster 3
**Cluster Variables**			
Stigma	0.51	0.55	−0.54
Violence	0.24	0.78	−0.39
PTSD	0.30	1.20	−0.57
Social support	−0.44	0.01	0.34
Resilience	−0.12	−0.43	0.20
PTG	0.16	0.40	−0.23
Sleep difficulty	−0.09	0.98	−0.23
**Gender Distribution**			
Male	74 (19.9%)	27 (7.3%)	118 (31.7%)
Females	62 (16.7)	26 (7%)	65 (17.5%)

## Data Availability

Data generated or analyzed during this study can be found at: https://doi.org/10.17632/4ggxwcbhdm.1.

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
