# Peer review of "Post-Traumatic Growth and Resilience among Hospitalized COVID-19 Survivors: A Gendered Analysis"

_ijerph, 2022, doi:10.3390/ijerph191610014_

Round 1

Reviewer 1 Report

Dear authors,

thank you for the interesting study! It is a well written manuscript which provides logical approach to the objectives, very detailed methods and appropriately presented results. The study also has relevant conclusions. Limitations are clearly explained. 

I have only very minor suggestions.
The first objective should start as : The first objective was...., 
Currently it reads as First, to investigate... which was confusing and I had to move to the second objective to understand that that was the first objective of the study.

Also in the line 132, I am not sure if the word "whereas" should be there or not. 

Overall, this was an interesting read and I have no further comments or concerns.

Author Response

We are very grateful for the comments. We have incorporated the suggestion into the revised manuscript.

Reviewer 2 Report

The theoretical basis and methodological contributions of a review are high and indicate that the quality of the paper meets the journal’s criteria for publication.
Despite the methodological shortcomings such as the cross-sectional design of the study and small study sample the results are important and bring new insight into existing knowledge.
The authors should carefully read and correct the introduction. In the introduction, it happens that the authors add a sentence not related to the previous scientific argument and without the correct citation source, which should not be found in scientific texts  (e.g. lines 64-5). Language and stylistic errors should be corrected to make the conclusions more clear for readers (e.g. lines 49-51) there are half-broken sentences in the text which must be corrected (e.g. lines 130-132).
The overall review of evidence linking the constructs to one another (Resilience and PTG)  is rather fragmented.
The authors used the modified versions of the original tools or only selected fragments of methods - with such changes they should check the psychometric values ​​of these scales - which was not done (e.g. EFA, CFA statistics)
Editorial errors should be corrected e.g. doubled spaces (line 142; 214; 378 and other lines), the absence of subtitle: Introduction; designation of p-value with an uppercase letter (line 341)

In experiencing PTG one of the important factors is time after a traumatic experience that significantly impacts transformational changes, this was not controlled by the authors, and at least some information (as study limitation) should be given

In the conclusion section, the authors mentioned the moderating role of gender – however, this was not examined in this study thus such a conclusion is not adequate to the results and must be corrected.

Author Response

Reviewer comment: The theoretical basis and methodological contributions of a review are high and indicate that the quality of the paper meets the journal’s criteria for publication. Despite the methodological shortcomings such as the cross-sectional design of the study and small study sample the results are important and bring new insight into existing knowledge.

Authors’ response: We are very grateful for the comments.

Reviewer comment: The authors should carefully read and correct the introduction. In the introduction, it happens that the authors add a sentence not related to the previous scientific argument and without the correct citation source, which should not be found in scientific texts (e.g. lines 64-5).

Authors’ response: Thank you very much for this feedback. We have reviewed the introduction section of the manuscript thoroughly and effected the necessary changes. It is our understanding that relevant literature has been cited and referenced.

Reviewer comment: Language and stylistic errors should be corrected to make the conclusions more clear for readers (e.g. lines 49-51) there are half-broken sentences in the text which must be corrected (e.g. lines 130-132).
Authors response: We have revised the manuscript, incorporating these recommendations.

Reviewer comment: The overall review of evidence linking the constructs to one another (Resilience and PTG)  is rather fragmented.
Author response: We have revised the manuscript accordingly.

Reviewer comment: Editorial errors should be corrected e.g. doubled spaces (line 142; 214; 378 and other lines), the absence of subtitle: Introduction; designation of p-value with an uppercase letter (line 341)
Author response: Thank you. These have been addressed.

Reviewer comment: In experiencing PTG one of the important factors is time after a traumatic experience that significantly impacts transformational changes, this was not controlled by the authors, and at least some information (as study limitation) should be given.

Author response: Thank you for the observation. We have noted this a study limitation. We responded as follows: “Time after a traumatic event is extremely important in PTG. The study findings relating to PTG are limited by the lack of inclusion time sensitive measures such as duration of treatment and after discharge.”

Reviewer comment: In the conclusion section, the authors mentioned the moderating role of gender – however, this was not examined in this study thus such a conclusion is not adequate to the results and must be corrected.

Authors response: This has been corrected by deleting the statement.

Author Response

We are grateful for the feedback. We have responded to the issues raised and revised the manuscript.

  1. We have provided information on how we calculated the overall prevalence in the text of Table 2.
  2. The correlation between PTG and resilience has been addressed. Likewise the no correlation witnessed for some of the study constructs have been mentioned.
  3. Providing individual correlations for the cluster was not helpful as we found no difference. In the interest of parsimony, we have agreed to keep the results in the current form.
  4. We have also responded to the minor issues.